The PARA-suite: PAR-CLIP specific sequence read simulation and processing

Kloetgen Andreas 1 2 3
Borkhardt Arndt 2
Hoell Jessica I. 2
McHardy Alice C. Alice.Mchardy@helmholtz-hzi.de AMC14@helmholtz-hzi.de 1 3
1 Department for Algorithmic Bioinformatics, Heinrich-Heine Universität Düsseldorf , Düsseldorf , Germany
2 Department of Pediatric Oncology, Hematology and Clinical Immunology, Medical Faculty, Heinrich-Heine Universität Düsseldorf , Düsseldorf , Germany
3 Computational Biology of Infection Research, Helmholtz Center for Infection Research , Braunschweig , Germany
Papaleo Elena
Electronic publication date: 2016 Oct 27
Publication date: 2016
Volume: 4
Electronic Location ID: e2619
Received 2016 May 25; Accepted 2016 Sep 27
Copyright: ©2016 Kloetgen et al.
Copyright year: 2016
Copyright holder: Kloetgen et al.
License: This is an open access article distributed under the terms of the Creative Commons Attribution License, which permits unrestricted use, distribution, reproduction and adaptation in any medium and for any purpose provided that it is properly attributed. For attribution, the original author(s), title, publication source (PeerJ) and either DOI or URL of the article must be cited.
License URL: https://creativecommons.org/licenses/by/4.0/

Keywords: Next-generation sequencing, Read alignment, Cross-linking and immunoprecipitation (CLIP), Read simulation, Posttranscriptional regulation, RNA-binding proteins

Funding: Comprehensive Cancer Center Düsseldorf/Deutsche Krebshilfe and the Medical Faculty of Heinrich Heine University Düsseldorf Elterninitiative Kinderkrebsklinik e.V. of Düsseldorf Helmholtz Centre for Infection Research Braunschweig This work was supported by the Düsseldorf School of Oncology (funded by the Comprehensive Cancer Center Düsseldorf/Deutsche Krebshilfe and the Medical Faculty HHU Düsseldorf). The authors additionally received funding from the Heinrich Heine University, the Elterninitiative Kinderkrebsklinik e.V. of Düsseldorf, and the Helmholtz Centre for Infection Research in Braunschweig. The funders had no role in study design, data collection and analysis, decision to publish, or preparation of the manuscript.

==============================
Background

Next-generation sequencing technologies have profoundly impacted biology over recent years. Experimental protocols, such as photoactivatable ribonucleoside-enhanced cross-linking and immunoprecipitation (PAR-CLIP), which identifies protein–RNA interactions on a genome-wide scale, commonly employ deep sequencing. With PAR-CLIP, the incorporation of photoactivatable nucleosides into nascent transcripts leads to high rates of specific nucleotide conversions during reverse transcription. So far, the specific properties of PAR-CLIP-derived sequencing reads have not been assessed in depth.

Methods

We here compared PAR-CLIP sequencing reads to regular transcriptome sequencing reads (RNA-Seq) to identify distinctive properties that are relevant for reference-based read alignment of PAR-CLIP datasets. We developed a set of freely available tools for PAR-CLIP data analysis, called the PAR-CLIP analyzer suite (PARA-suite). The PARA-suite includes error model inference, PAR-CLIP read simulation based on PAR-CLIP specific properties, a full read alignment pipeline with a modified Burrows–Wheeler Aligner algorithm and CLIP read clustering for binding site detection.

Results

We show that differences in the error profiles of PAR-CLIP reads relative to regular transcriptome sequencing reads (RNA-Seq) make a distinct processing advantageous. We examine the alignment accuracy of commonly applied read aligners on 10 simulated PAR-CLIP datasets using different parameter settings and identified the most accurate setup among those read aligners. We demonstrate the performance of the PARA-suite in conjunction with different binding site detection algorithms on several real PAR-CLIP and HITS-CLIP datasets. Our processing pipeline allowed the improvement of both alignment and binding site detection accuracy.

Availability

The PARA-suite toolkit and the PARA-suite aligner are available at https://github.com/akloetgen/PARA-suite and https://github.com/akloetgen/PARA-suite_aligner, respectively, under the GNU GPLv3 license.

Background

RNAs play a crucial role in cell survival and viability. Coding messenger RNAs (mRNAs), which are translated into proteins, and many other RNA species, such as small and long non-coding RNAs, ribosomal RNAs and transfer RNAs, are essential for the survival and proper functioning of the cells (Eddy, 2001). Most RNAs maintain their function by working together with the so-called RNA-binding proteins (RBPs) (Glisovic et al., 2008). RBPs are involved in virtually all steps of the mRNA lifecycle, from polyadenylation, translocation and modification to translation (Hieronymus & Silver, 2004). Thus, it is not surprising that many RBPs that show aberrant functions or changes in expression patterns have been associated with disease progression or even with carcinogenesis (Lukong et al., 2008). For instance, the FET protein family, which consists of the three RBPs FUS, EWSR1 and TAF15, is ubiquitously expressed and widely conserved in mammals. Genomic rearrangements, leading to mutant forms of these RBPs in humans, have been described as key players in sarcomas and leukemia (Tan & Manley, 2009). More recently, two mutants of FUS causing amyotrophic lateral sclerosis have shown different RNA-binding patterns compared to their wild-type counterparts, supporting the importance of the function of FUS in mRNA processing (Hoell et al., 2011).

Experimental protocols have been developed to analyze the functional network in which a particular RBP interacts. A promising method for this purpose is the photoactivatable ribonucleoside-enhanced cross-linking and immunoprecipitation (PAR-CLIP) technique (Hafner et al., 2010). When coupled with deep sequencing, it identifies the bound RNAs for a particular RBP on a genome-wide scale. First, the cells are supplied with a specific photoactivatable nucleoside, such as 4-thiouridine (4-SU), which is incorporated as an alternative to the respective nucleoside into nascent mRNA transcripts. Afterwards, the cells are treated with ultraviolet light at 365 nm to cross-link the amino acids of RBPs to the nucleotides of their bound RNA molecules. The incorporation of 4-SU instead of uridine results in nucleotide conversions from uridine to cytidine at all cross-linked sites containing a 4-SU during reverse transcription (a necessary step for preparing cDNA libraries for sequencing). This specific replacement is called a ‘T–C conversion’. T–C conversions can be used to distinguish between non-specifically bound RNA fragments (considered as contaminations) and those that are specifically bound and cross-linked to the RBP of interest (Ascano et al., 2012a; Golumbeanu, Mohammadi & Beerenwinkel, 2015). We recently published a detailed protocol for the PAR-CLIP procedure (Hoell et al., 2014). Other CLIP protocols for the genome-wide identification of RBP targets are also frequently used, such as high-throughput sequencing of RNAs isolated by cross-linking and immunoprecipitation (HITS-CLIP, sometimes also called CLIP-seq) or the iCLIP protocol (Chi et al., 2009; König et al., 2010). The procedures, experimental designs and bioinformatic analysis of these different CLIP methods differ greatly and are still evolving. Recent reviews compare the strengths and weaknesses of the three methods in detail (Wang et al., 2015; Danan, Manickavel & Hafner, 2016). HITS-CLIP, for example, mainly introduces deletions of a single base at the cross-linked sites, whereas single nucleotide conversions do not seem to occur at a significant frequency (Zhang & Darnell, 2011; Sugimoto et al., 2012).

Current sequencing platforms allow for the sequencing of mammalian transcriptome libraries with high coverage. Nowadays, the most commonly used next-generation sequencing (NGS) platforms are 454, Illumina, IonTorrent and PacBio (Van Dijk et al., 2014). Depending on the sequencing platform and the sample type, sequencing errors vary in type and frequency. The errors that most commonly occur are substitution errors and indels of a few bases between the sequencing read and the reference sequence (large rearrangements, such as those leading to chimeras, are also possible errors but are not discussed here) (Laehnemann, Borkhardt & McHardy, 2015). In an RNA-Seq dataset, a single transcript will be covered by sequencing reads in all its expressed coding exons (apart from, for example, amplification errors or alternative splicing variants). For common sequencing data types, such as RNA-Seq and DNA-Seq, designated read aligners have recently been developed. These include short read aligners, such as BWA (Li & Durbin, 2009) or Bowtie (Langmead et al., 2009), and read aligners such as TopHat (Trapnell, Pachter & Salzberg, 2009), STAR (Dobin et al., 2013) or Subjunc (Liao, Smyth & Shi, 2013), which can also handle longer sequencing reads spanning exon–exon junctions. Specific software for the evaluation and analysis of the PAR- and HITS-CLIP sequencing data is needed to accommodate their unique error profiles (Kloetgen et al., 2015). For instance, the read aligner BWA PSSM (Kerpedjiev et al., 2014) makes use of a pre-defined position-specific scoring matrix to process the error-prone PAR-CLIP reads.

In general, the sequencing error profiles of RNA-Seq datasets, including PAR-CLIP data, can vary between different sequencing runs, depending on the sequencing machine, the experimental conditions and the biological properties of the sample (Laehnemann, Borkhardt & McHardy, 2015; Schirmer et al., 2015). Here, we describe the PAR-CLIP analyzer suite (PARA-suite), which includes a PAR-CLIP read simulator, an error estimation tool for CLIP datasets and an alignment pipeline based on a novel alignment algorithm performing on-the-fly dataset-specific error estimation. The alignment pipeline thus automatically adjusts to the quality and error profiles of individual sequencing datasets. We compare PAR-CLIP sequencing reads to regular transcriptome sequencing reads (RNA-Seq) to identify the distinctive properties that are relevant for reference-based read alignment and RBP binding site detection from PAR-CLIP datasets. Generation of simulated PAR-CLIP datasets can be performed with the PARA-suite’s read simulator. The PARA-suite toolkit is available at https://github.com/akloetgen/PARA-suite and https://github.com/akloetgen/PARA-suite_aligner, implemented as an extension of BWA (henceforth referred to as BWA PARA). It is licensed under GNU GPLv3, and can be implemented in the programming languages Java and C.

Methods

Datasets and read aligners

We downloaded PAR-CLIP data for the FET family (EWSR1, FUS and TAF15) from the DRASearch database (https://trace.ddbj.nig.ac.jp/DRASearch/) with the accession number SRA025082 (Hoell et al., 2011), the HuRdataset with the accession number SRR248532, the MOV10dataset with the accession number SRR490650 and the HITS-CLIP data on the Argonaute2 protein (AGO2) (Chi et al., 2009) from http://ago.rockefeller.edu/. For estimating the error profiles of regular RNA-Seq runs, we downloaded two sequencing lanes from an NGS quality assessment study with the accession numbers SRR896663 and SRR896664 (SEQC/MAQC-III-Consortium, 2014) from DRASearch and pooled the data. An overview of the analyzed datasets can be found in Table 1.

Table 1 Overview of the analyzed RNA-Seq and CLIP datasets.

Dataset	Published (year)	Sequencing method	Platform	Accession number/website	
EWSR1	2011	PAR-CLIP	Illumina Genome Analyzer II	SRA025082	
FUS	2011	PAR-CLIP	Illumina Genome Analyzer II	SRA025082	
TAF15	2011	PAR-CLIP	Illumina Genome Analyzer II	SRA025082	
HuR	2011	PAR-CLIP	Illumina Genome Analyzer	SRR248532	
MOV10	2012	PAR-CLIP	Illumina Genome Analyzer II	SRR490650	
AGO2	2009	HITS-CLIP	Illumina Genome Analyzer II	http://ago.rockefeller.edu/	
Human reference RNA	2014	RNA-Seq	Illumina HiSeq 2000	SRR896663, SRR896664	

We used the following read aligners and versions, shown in alphabetic order: Bowtie, version 0.12.7 (Langmead et al., 2009), Bowtie2, version 2.2.3 (Langmead & Salzberg, 2012), BWA, version 0.7.8 (Li & Durbin, 2009), BWA PSSM, initial release version (Kerpedjiev et al., 2014), MOSAIK, version 2.2.3 (Lee et al., 2014), STAR, version 2.3.0 (Dobin et al., 2013), Subjunc, version 1.4.2 (Liao, Smyth & Shi, 2013) and TopHat, version 2.0.13 (Trapnell, Pachter & Salzberg, 2009).

PAR-CLIP read simulator and hierarchical clustering

We developed a PAR-CLIP read simulator (Fig. 1) that creates short RNA reads which mimic important PAR-CLIP specific properties ‘Properties of PAR-CLIP reads’. First, the following probability distributions are obtained from real PAR-CLIP data: (a) a probability matrix ε representing the background error profile of sequencing errors, (b) a probability vector of T–C conversion frequencies α for ranked T–C conversion sites, (c) a probability vector β for the preferred read positions of T–C conversion sites within binding sites, (d) a probability vector μ for indel frequencies per read position and (e) a probability vector δ for the base-calling quality score distribution per read position. The probability matrix ε contains a probability distribution for each DNA base over the DNA bases {A, C, G, T}. For this purpose, a PAR-CLIP dataset is aligned against a reference genome sequence with an appropriate read aligner.

Figure 1 Pipeline of the PAR-CLIP read simulator implemented in the PARA-suite.

Part A describes the process of generating the error profile and other parameters learned from a real PAR-CLIP dataset. Part B starts to generate reads mapping to RBP binding sites (clusters) on transcript regions from a given transcript database (e.g., Ensembl genes). In Part C, the pre-calculated profiles are used to introduce T–C conversions, sequencing errors, indels and base-calling quality scores to the defined reads.

Based on these alignments, the sequencing error profile ε is estimated from the observed frequencies of all single nucleotide substitutions, except for T–C errors, as these include PAR-CLIP specific T–C conversions. Standard T–C sequencing errors are approximated by the average over all the other sequencing error frequencies. The probability vectors 00B5 and δ are also inferred from these alignments. Next, all aligned reads of the real dataset are clustered (stacked) using single-linkage hierarchical clustering based on their genomic mapping positions, using a 5-base overlap of the genomic mapping positions as the clustering threshold. To identify high confidence clusters (sometimes referred to as binding sites) as defined in the literature (Hafner et al., 2010), clusters that contain less than 10 reads, less than 25% T–C conversions per cluster, are longer than 75 bases and include only T–C conversion sites that are reported as single nucleotide polymorphism loci in the dbSNP database (version 142) (Sherry et al., 2001) are discarded. This implementation of hierarchical clustering is part of the PARA-suite and will later be used for binding site detection. For the subsequent simulation, the positions and frequencies of highly mutated T–C sites within reads are determined to estimate α and β from the high confidence clusters (Figs. S1A–S1B).

Next, the PAR-CLIP read simulation starts with the random selection of transcripts from a pre-selected database of annotated transcripts. One to at most three clusters (the number of clusters is randomly chosen from a uniform distribution) containing several reads are created for a selected transcript sequence. The starting positions of the clusters are randomly selected from a uniform distribution within the entire range of a transcript. The number of reads simulated for a single cluster is drawn from a normal distribution with a mean of 16 and a standard deviation of 10. This enables the simulation of a wide range of read coverages throughout the clusters. Furthermore, small shifts of the start and end site of each read leading to distinctive alignment position shifts in the shape of a cluster are randomly introduced at this step (normal distribution, standard deviation = 1). A user-defined parameter λ ∈ [0,1] specifies the fraction of clusters that are considered to be binding-sites, whereas the remaining clusters mimic contaminations of unbound RNAs that occur in all PAR-CLIP experiments. We recommend values in the range of 0.5–0.7 (50–70%), as we observed this range of aligned sequencing reads stacking into clusters after hierarchical clustering and filtering (Table S1; similar values were previously reported by Ascano et al. (2012a)). If more than one T–C site is simulated for a single cluster, a major T–C conversion site is selected according to the site-specific T–C conversion profile β and T–C conversion probabilities are drawn from α. Subsequently, background sequencing errors are introduced on the basis of the pre-computed probability matrix ε and the frequency vector μ for substitutions and indels, respectively. In the last step, every base receives a base-calling quality score, as specified by the position-specific quality score distribution δ. All generated reads are stored in the universal FASTQ format (Cock et al., 2010). The PAR-CLIP read simulator is available through the PARA-suite.

The PARA-suite: tools for error profile inference, read simulation, multiple database mapping and more

The PARA-suite is a toolkit for processing and aligning short and error-prone sequencing reads. It is implemented in Java using HTSjdk, a Java API for high-throughput sequencing data formats (https://github.com/samtools/htsjdk). The PARA-suite allows the user to estimate a sequencing run-specific error profile, combine the results of multiple reference database alignments, cluster an aligned sequencing read dataset (‘PAR-CLIP read simulator and hierarchical clustering’), run the PAR-CLIP read simulator, benchmark an alignment of simulated PAR-CLIP sequencing reads and run a full processing pipeline for error-prone short read alignments (Fig. 2A). The alignment pipeline of the PARA-suite includes the calculation of an error profile for a particular sequencing run, applying the alignment algorithm described in the following section, and optionally combines the results of read mappings against multiple databases (Figs. 2B–2D). First, a read alignment against a reference sequence is performed with a fast short read aligner. By default, this is carried out with BWA, as our evaluations have demonstrated this to be a fast and accurate aligner (‘Accuracy of common read aligners and the PARA-suite on simulated PAR-CLIP 403 data’) on PAR-CLIP reads. However, other read aligners can also be used to produce the reference-based read alignment. This initial read alignment is used to estimate the underlying mismatch and indel probabilities M, I and D (as described in the next section) of the sequencing run. Once the error profile has been estimated, all sequencing reads can be aligned with BWA PARA (‘Algorithm of the PARA-suite aligner BWA PARA’) against the reference sequence(s). All aligned reads are reported in a BAM file.

Figure 2 The PARA-suite.

(A) The PARA-suite. Dashed boxes represent software packages; all other boxes represent executable programs. The Utils package includes tools for working with error-prone sequencing data and the postprocessing package contains a tool for clustering an aligned PAR-CLIP dataset to identify RBP-bound genomic regions. (B) Read alignment by a fast read aligner is necessary to infer the error profile for a particular read dataset (we selected BWA). (C) BWA PARA is applied to the entire dataset to map error-prone reads, indicated here by the additional mapping of the two reads (shown in blue). (D) An optional alignment versus a transcriptome reference database can be executed using BWA PARA to identify previously unmapped reads.

Algorithm of the PARA-suite aligner BWA PARA

The general BWA algorithm uses a Burrows–Wheeler transform (BWT) (Burrows & Wheeler, 1994) to create an index for a reference genome sequence and applies a backward search to identify possible mapping positions in the genome for every single sequencing read. The backward search starts with the last base of a read proceeding to its front, searching the partly decompressed suffix trie using the auxiliary Ferragina and Manzini index (Ferragina & Manzini, 2000) for a matching predecessor base of the read’s bases compared so far. Even if a match can be found for a single comparison, mismatches are introduced and all possible downstream paths within the suffix trie are considered until a pre-defined threshold of maximal mismatches is exceeded in a single path (Fig. 3, red dotted line).

Figure 3 Suffix trie paths for the BWA and PARA-suite.

Paths of the algorithms through the suffix trie aligning the read sequence GCCATG$ against the reference sequence GTTATG$ (where $ means the end of a sequence). The red dotted line represents the algorithm of the BWA aligner, allowing for two mismatches; the blue dashed line indicates the BWA PARA algorithm. The underlined bases represent positions where the respective aligner introduces a mismatch. The example shows that BWA PARA needs 14 comparisons but the basic BWA needs 16 comparisons. Indels are not shown for simplicity.

The principal idea of BWA PARA is the introduction of a probability estimate for each comparison of the backward search. This enables mismatches to be weighted according to their probabilities that they occur in the analyzed dataset. A sequencing run is initially characterized according to its underlying error probabilities. This allows us to determine specific error-profiles for experimental techniques, such as the frequent T–C conversions in PAR-CLIP data, which are more common than sequencing errors. The error profile M is a 4 × 4 probability matrix specifying substitution probabilities values ∈ [0..1] for each reference base ∈ {A, C, G, T} to the read bases {A, C, G, T} (Fig. 4A). Indels are introduced during the alignment step separately, using the estimated probabilities I ∈ [0,1] for insertions and D ∈ [0,1] for deletions.

Figure 4 The BWA PARA alignment approach.

(A) The error profile probability matrix M and the indel probabilities I and D, which are used as input for the BWA PARA algorithm, as well as exemplary results of the intermediate calculations of the BWA PARA algorithm. In M, only T–C conversions have a higher probability (6.3%) than sequencing errors and indels. (B) The last characters of a particular read and three examples of mapping positions within a reference, called ref a–c. (C) The calculation of a maximum threshold T for the mapping probability p (see the Equation 2 in the main text, and values from (A) in this image). (D) The mapping probability calculation of the read when mapped to References a–c. The read fails to map against ref b with two sequencing errors, whereas ref a and ref c are suitable mapping positions, where the probability p is higher than the threshold T. For implementation, we worked with the open-source read aligner BWA (version 0.7.8) to extend its algorithm for the alignment of short and error-prone reads.

For each comparison between a read base at read position i (read[i]) and a reference base at position j (ref[j]) in the reference sequence, the algorithm recursively calculates a joint probability value p, which is used to examine the chance of incorporating a matching base or a suitable error, including indels, at the respective read positions (Fig. 4D): pi=pi+1⋅D,ifrefjisdeletedpi+1⋅I,ifreadiisinsertedpi+1⋅Mreadi,refj,otherwise

with p|read| =1, starting with i = |read| –1 and decreasing i at each step, except in the case of a deletion (where i is left unchanged), for i ≥ 0.

Before the alignment of a particular read, a minimal threshold T for the probability p is needed to decide whether a read is accepted as aligned or rejected. The calculation for T depends on a parameter X for the average number of mismatches. Note that this is not a maximal threshold in terms of absolute mismatches, as the number of the more frequent errors per aligned read can exceed X. The parameter X can be pre-defined by the user or is by default estimated as the expected number of mismatches for different read lengths based on the error profile M for a sequencing run. Next, the minimal threshold T is computed (Figs. 4B and 4C): T=avgmatch|read|−X⋅avgmismatchX,

where avgmatch=15∑i∈0..3Mi,i+1−1+Dandavgmismatch=114∑i,j∈0..3;i≠jMi,j+I+D.

Both avg(match) and avg(mismatch) are normalized by the number of elements (four matches plus one for no indel occurring, and 12 mismatches plus 2 for either a insertion or a deletion). If p falls below the pre-calculated threshold T during read alignment, the path within the suffix trie is assumed not to match the read and is rejected (Fig. 3, blue dashed line). The algorithm thus penalizes rare types of mismatches according to M, whereas frequent errors, such as T–C errors in PAR-CLIP reads, are the most favored substitutions in the alignment process (Figs. 4B–4D).

Results

Properties of PAR-CLIP reads

To assess the most important properties of the PAR-CLIP sequencing reads for read alignment, we systematically compared PAR-CLIP datasets for the three RBPs EWSR1, FUS and TAF15 (the FET protein family) (Hoell et al., 2011) to a recently published RNA-Seq run on human reference RNA (SEQC/MAQC-III-Consortium, 2014). The 10 outermost bases of the SEQC/MAQC reads showed error rates with peaks at 1.5 and 2.2 errors per 100 reads (EPR). In contrast, the middle read length range showed an average of about 0.3 EPR (Fig. S2A, red line). As the short reads of the FET PAR-CLIP datasets consisted only of these outermost bases, they exhibited a two- to threefold higher average sequencing error rate (about 0.7 EPR or even higher) than the SEQC/MAQC reads (Fig. S2B, green line). When considering the T–C conversions only, we observed 1.319 EPR for EWSR1, 1.477 EPR for FUS and 1.051 EPR for TAF15 on average. This is an approximately 20- to 30-fold increase in comparison to the SEQC/MAQC dataset with 0.051 EPR for T–C conversions on average (Fig. S2). Moreover, we analyzed data from two further PAR-CLIP studies performed on the RBPs HuR (Mukherjee et al., 2011) and MOV10 (Sievers et al., 2012), which showed similar error profiles and EPRs to the FET PAR-CLIPs for T–C conversions (Fig. S3).

Further analyses of the PAR-CLIP read datasets for EWSR1, FUS, TAF15, MOV10 and HuR showed the PAR-CLIP reads (a) to be shorter than 30 bases, (b) to cover only short stretches of an expressed gene rather than the entire expressed RNA (these stretches are henceforth called clusters), (c) to exhibit a specific nucleotide conversion pattern with a strong enrichment of T–C conversions, where (d) such conversions occur in specific ‘conversion sites’ in the clusters. The two properties (a) and (b) are determined by treating the cells with RNAse T1 or the lysate during the PAR-CLIP experimental protocol. As only short RNA fragments that are not digested by the endonuclease (these are probably protected by the binding pocket of the RBP) are sequenced, the lengths of those fragments are usually short. However, the nucleotide composition of those reads is strongly affected by the digestion enzyme and can vary among different digestion enzymes (Kishore et al., 2011). After quality trimming and adapter trimming of the five PAR-CLIP datasets, the average read lengths were 25.67 bases (EWSR1), 25.60 bases (FUS), 24.21 bases (TAF15), 25.20 bases (HuR) and 23.36 bases (MOV10). As the transcript regions outside the bound RNA fragment are digested by the endonuclease, these are removed during immunoprecipitation and not sequenced, except for additional binding sites on the same transcript further up- or downstream. Thus, the sequencing reads are stacked into short clusters covering short stretches of the gene and representing the RBP-bound regions of the transcripts (Fig. S4A).

The two properties (c) and (d) were determined by incorporating photoactivatable nucleosides into the nascent transcripts during transcription. In the case of 4-SU, T–C conversions occur in the sequencing reads at all cross-linked sites, where the 4-SU is incorporated instead of the native uridine. These conversions can reach high rates in specific conversion sites within a cluster (Hafner et al., 2010). In the analyzed datasets, we observed an average frequency of about 70% T–C conversions in the main T–C conversion site (Fig. S1A). This emphasizes that simulated read datasets with specific properties are necessary for the evaluation of common short read aligners for analyzing PAR-CLIP read data. However, this cannot be created by common sequencing read simulators, such as ART (Huang et al., 2012) or GemSIM (McElroy, Luciani & Thomas, 2012). These produce simulated reads with a continuous coverage over the entire transcript range and the introduced mutations are distributed randomly throughout the simulated reads. This is not the case for PAR-CLIP sequencing reads.

PAR-CLIP read simulation for performance evaluation

We simulated a total of 10 PAR-CLIP read datasets based on information learned from three previously published PAR-CLIP datasets of the FET protein family (Hoell et al., 2011) (Table S2). We imitated Illumina GenomeAnalyzer II sequence data according to the real datasets used. The respective sequencing error and T–C conversion profiles were generated on the basis of alignments of all three datasets against the human reference genome sequence version 38 (GRCh38) (Lander et al., 2001). The error profile and additionally estimated distributions were similar to the ones from PAR-CLIP data on the two RBPs HuR and MOV10, indicating that these profiles represented a reasonable approximation for PAR-CLIP data in general. We selected human transcript sequences downloaded from Ensembl Genes version 77 (Cunningham et al., 2015) as our sequence database to simulate human transcript read sequences. We set λ, the parameter for the fraction of sequencing reads that stacked into clusters bound by the RBP, to 65%. These true RBP binding sites showed high T–C conversion frequencies in different T–C conversion sites. The remaining 35% of the simulated sequencing reads were designated to represent non-specifically bound transcripts without an elevated T–C conversion rate, except for a few T–C sequencing errors. These reflected RNA contaminations that can occur during the PAR-CLIP experiment.

To assess the quality of the simulation, we then compared PAR-CLIP-specific properties between the 10 simulated datasets and the FET PAR-CLIP data. Within a cluster detected in a simulated dataset, shifts in the alignment positions of a few nucleotides at the beginning and the end of the simulated cluster could be seen between the reads (Fig. S4B). According to the position-wise T–C conversion profile used, a T–C conversion site with a high conversion rate, as well as a few sites with lower conversion rates, were usually present in the detected clusters (e.g., Fig. 1B). We compared the error profiles between one of the simulated datasets and the real datasets, and distinguished between T–C errors and all other errors; the latter represent all sequencing errors other than the T–C sequencing errors (Fig. S2C). Similar to the real data, the distribution of the sequencing errors in the simulated dataset peaked at the beginning of the reads and dropped to a mean error rate of 0.6 EPR in the middle read length range. Error rates were slightly underestimated in the simulated data compared to the real PAR-CLIP data, presumably because of a small percentage of multiple mutations that occurred at individual sites. Apart from this, the simulated datasets appeared to be representative of real PAR-CLIP data in the relevant aspects.

Accuracy of common read aligners and the PARA-suite on simulated PAR-CLIP data

Using the simulated PAR-CLIP datasets, we analyzed the accuracy of state-of-the-art read aligners and common binding site detection algorithms, and compared these to the PARA-suite alignment pipeline. The aligners BWA and Bowtie have often been used in CLIP studies (Lebedeva et al., 2011; Ascano et al., 2012b; Sievers et al., 2012). BWA PSSM was applied with the PSSM for PAR-CLIP provided by its authors because a PSSM estimated from the sequencing dataset revealed worse accuracy (data not shown). MOSAIK was executed, reporting only unique mappings, allowing for up to three mismatches between the read and the reference sequence, and using a Smith–Waterman bandwidth of 5. The read aligners were used to align the simulated datasets to the reference sequence GRCh38. We also executed the PARA-suite on the Ensembl Genes transcriptome database (version 77) and combined the results with the genomic reference sequence alignments. These results are henceforth referred to as those of the “PARA-suite pipeline”, whereas the results of the genomic alignment step using the PARA-suite only are referred to as those of “BWA PARA.” For BWA PARA, the sequencing error and T–C conversion profiles for the simulated datasets were obtained on the basis of the BWA alignments, allowing for two mismatches (BWA 2MMs) for each of the simulated datasets separately (execution commands are outlined in the Supplemental Information 1). For an overview of the performance, we estimated the average of the recall, precision and accuracy for each aligner over the 10 simulated datasets (our calculations are described in the Supplemental Information 1). Unfortunately, BMix does not report negative clusters (contaminations) and thus we were able to neither calculate the recall nor the accuracy, but only the precision.

In terms of overall performance, the PARA-suite performed best, with an accuracy of 69.74% for BWA PARA and 73.14% for the entire pipeline, showing performance gains of 1.57% and 4.97% compared to the second-best aligner (BWA 2MM), respectively (Table 2, Table S3). Many prominent PAR-CLIP studies have used Bowtie 1MM or BWA 2MM for the read alignment step (Lebedeva et al., 2011; Mukherjee et al., 2011; Ascano et al., 2012b; Sievers et al., 2012; Mukherjee et al., 2014). When we compared the PARA-suite pipeline with these two aligners, the PARA-suite pipeline showed an increase of 16.95% and 4.97% in the overall accuracy, respectively. Notably, 1.56% of the reads aligned by the PARA-suite pipeline on average spanned an exon–exon junction. These were not identified by the genomic reference mapping step but instead required alignment against the transcriptome reference sequences. Additionally, we compared the recall (the fraction of correctly aligned reads out of all simulated reads) and the precision (the fraction of correctly aligned reads out of all aligned reads) to assess the mapping ability of the read aligners (Table 2, Fig. S5). Here, the PARA-suite pipeline and BWA PARA were ranked first and third regarding recall, and first and second regarding precision, respectively, out of 10 analyzed alignment scenarios (Table 2). Hence, the PARA-suite pipeline and BWA PARA offer notable performance increases over commonly applied alignment setups.

Table 2 Alignment accuracy on simulated PAR-CLIP data.

The most accurate alignment results were obtained for different parameter settings for each read aligner on 10 simulated PAR-CLIP datasets. The results are averaged per read aligner over all 10 datasets and are sorted by accuracy.

Aligner	Accuracy (in %)	Variance	Recall (in %)	Precision (in %)	Mapped overall	Mapped correctly	Real time (s)	
PARA-suite pipeline	73.14	1.37E–06	84.49	71.85	1,024,792	969,948	396.8	
BWA PARA	69.74	1.38E–06	82.16	68.24	975,672	924,802	153.7	
BWA 2MMs	68.17	1.37E–06	82.31	64.98	959,171	904,034	359.2	
Bowtie 2MMs	63.38	1.10E–06	77.91	60.93	886,512	840,540	120.6	
BWA PSSM	59.80	1.18E–06	74.04	58.72	818,895	793,007	25.4	
TopHat	59.69	8.35E–07	76.10	55.35	844,902	791,549	282.9	
Bowtie2	56.22	1.11E–06	73.23	51.43	763,893	745,531	13.4	
STAR	50.74	9.10E–07	69.57	43.02	826,871	672,920	248.6	
MOSAIK	44.88	2.18E–04	62.83	37.16	897,679	595,220	12,128.18	
Subjunc	35.42	9.03E–07	50.61	26.09	597,400	469,751	64.2	

Table 3 Binding sites detected for the FET protein family.

The number of binding sites for the FET protein family identified by the aligners BWA 2MM, BWA PSSM, BWA PARA, Bowtie 2MM and Bowite2 in combination with BMix and the hierarchical clustering of the PARA-suite. Filters were applied according to ‘PAR-CLIP read simulator and hierarchical clustering.’

	EWSR1	FUS	TAF15	
BWA 2MM BMix	20,703	14,768	5,086	
BWA 2MM Clustering	22,760	36,861	5,810	
BWA PSSM BMix	24,639	19,628	5,238	
BWA PSSM Clustering	27,550	51,606	6,130	
BWA PARA BMix	25,478	19,006	5,862	
BWA PARA Clustering	28,692	48,042	7,588	
Bowtie 2MM BMix	19,173	13,902	4,582	
Bowtie 2MM Clustering	21,082	35,490	5,254	
Bowtie2 BMix	12,384	8,078	3,558	
Bowtie2 Clustering	13,338	20,398	3,710	

We then tested the accuracy of the binding site detection algorithms BMix, PARalyzer and the hierarchical clustering of the PARA-suite using the read alignments of BWA PARA (Table S4). The hierarchical clustering identified the most correct binding sites: 3.26% more correct sites than BMix and 5.54% more correct binding sites than PARalyzer. However, BMix identified fewer false binding sites than the hierarchical clustering (20.30% fewer) and PARalyzer (69.85% fewer). Furthermore, we investigated whether BWA PARA increased the number of binding sites detected, irrespective of the detection algorithm used. In conjunction with BMix, BWA 2MM (the second-best aligner) identified 7.17% fewer correct binding sites than BWA PARA. With PARalyzer, BWA 2MM identified 2.97% fewer correct binding sites than BWA PARA. Finally, the hierarchical clustering identified 7.52% more correct binding sites for BWA PARA than for BWA 2MM. Overall, the combination of BMix and BWA PARA provided the most accurate results on our simulated data.

Analysis of FET PAR-CLIP datasets

To investigate the performance of the PARA-suite on real PAR-CLIP datasets, we applied it to the three FET PAR-CLIP datasets (Hoell et al., 2011). The sequencing reads were preprocessed similarly to the method given in the original publication, and low quality ends and adapter sequences were trimmed using Cutadapt (Martin, 2011). Afterwards, all remaining reads longer than 18 bases were aligned against GRCh38 with Bowtie2, Bowtie 2MM, BWA 2MMs, BWA PSSM and BWA PARA (without the transcriptome mapping step to achieve comparable results). Selection of the read aligners (i.e., Bowtie2, Bowtie 2MM, BWA 2MM, BWA PSSM and BWA PARA) was based on the results of the previous section, as these represent the most accurate read aligners on PAR-CLIP data. We measured the fraction of aligned reads for all the aligners on the three datasets (Table S5). BWA PARA generated the largest fraction of aligned reads over all three datasets in comparison to BWA 2MM and BWA PSSM. Next, we stacked (clustered) all the aligned reads using BMix and the hierarchical clustering tool of the PARA-suite (Table 3). BWA 2MM identified fewer binding sites than BWA PSSM or BWA PARA for read alignments prior to either BMix or hierarchical clustering. Using the hierarchical clustering, BWA PARA reported the largest number of binding sites for two out of the three datasets. BWA PSSM identified 6.90% more clusters than BWA PARA for the FUS dataset whereas BWA PARA identified 3.98% more clusters for the EWSR1 dataset and 19.21% more clusters for the TAF15 dataset than BWA PSSM. In comparison to the values reported in the original publication, the use of BWA PARA and hierarchical clustering increased the number of binding sites by 33.71% for EWSR1 and 16.77% for FUS, and decreased them by 12.56% for TAF15. After extracting distinct genes from all binding sites identified by BWA PARA (10,631 genes in total), 26.90% additional genes were found for all three RBPs, in comparison to the original publication (7,771 genes in total). As expected for three RBPs from the same family, there was a substantial overlap in terms of the identified genes, with 2,702 genes targeted by all three RBPs (Fig. S6).

Analysis of PAR-CLIP data on HuR

We next applied the PARA-suite to a PAR-CLIP dataset on HuR, an RBP promoting RNA stabilization (Mukherjee et al., 2011). Adapters and low-quality ends within the HuR dataset were trimmed using Cutadapt and reads shorter than 14 bases were discarded. The binding motif of HuR is well-studied and is AU-rich, with a consensus motif described as AUUUA, AUUUUA or AUUUUUA (Nabors et al., 2003; Lebedeva et al., 2011), showing potentially more T–C conversions within each binding site than other RBPs. As the generated error profile of the dataset was similar to those of the FET PAR-CLIP data (‘Properties of PAR-CLIP reads’), the data quality seemed comparable. However, we noted a slight increase in T–C conversions (Fig. S3). The AU-rich binding motif might explain the higher T–C conversion rate of 1.684 EPR compared to the conversion rate of 1.477 EPR e.g., for FUS.

We used the same read aligners as described in the previous section (Bowtie2, Bowtie 2MM, BWA 2MM, BWA PSSM and BWA PARA) to align the pre-processed dataset against the human genome reference GRCh38. We applied BMix and the hierarchical clustering of the PARA-suite to determine the binding sites of HuR derived by using the different read aligners. BWA PSSM, in conjunction with BMix, identified the most RBP binding sites within the genome, which was 3.69% more than BWA PARA (Table 4). When we compared the binding sites detected by BMix and the PARA-suite hierarchical clustering for alignments created by BWA PARA (binding site positions overlapping by at least 13 bases), the difference was only marginal, with an overlap of more than 98.25% for the two methods. A recent study of this dataset reported binding sites using Bowtie 2MM for the alignment step and PARalyzer for the binding site detection. We found that the use of either BWA PSSM or BWA PARA in conjunction with either BMix or hierarchical clustering increased the number of binding sites detected by 2.87–7.84%.

Table 4 Binding sites detected for HuR.

Binding sites detected by BMix and the hierarchical clustering based on read alignments performed by BWA 2MM, BWA PSSM, BWA PARA, Bowtie 2MM and Bowtie2 on the HuR dataset.

	BMix	Hierarchical clustering	
BWA 2MM	136,775	137,697	
BWA PSSM	147,883	148,985	
BWA PARA	141,365	141,867	
Bowtie 2MM	125,592	125,067	
Bowtie2	88,369	87,400	

We searched for the exact binding motifs of HuR (ATTTA, ATTTTA and ATTTTTA) within the binding sites detected by BMix within 3′ untranslated region (UTR) or introns for all the read aligners tested. We found that all aligners performed comparably, with motifs present in 42–44% of all binding sites detected. The largest fraction was achieved using read alignments with BWA PSSM (44.33%), whereas BWA PARA in combination with BMix found 42.53% of the binding sites that were most likely correct. Bowtie 2MM in combination with BMix had the lowest fraction of binding sites containing the reported binding motif (42.44%). We also compared the previously reported HuR binding sites to the binding sites determined by the full PARA-suite pipeline with BMix for clustering and detected 13 out of 15 sites, namely 3′ UTR PTGS2, 3′ UTR CDKN1A, 3′ UTR VEGFA, 3′ UTR TNF, 3′ UTR SLC7A1, 3′ UTR CCND1, 3′ UTR MYC, 3′ UTR XIAP, 3′ UTR CELF1, TTS CSF2, 3′ UTR CCNB1, intron NCL and 3′ UTR KRAS. The binding information for this comparison was taken from the Ingenuity knowledge base (Calvano et al., 2005). The original study on the HuR dataset (Mukherjee et al., 2011) only reported 12 out of these 15 genes having confirmed binding site.

Discussion

We provided a detailed characterization of the error profiles of PAR-CLIP reads and an in-depth performance assessment of short read aligners in combination with binding site detection tools. We characterized some of the unique properties of PAR-CLIP sequence datasets, including the preferred read positions for T–C conversion sites and their frequencies per read position. We observed higher frequencies of sequencing errors in PAR-CLIP data than in the human reference RNA-Seq data. A likely reason for this behavior could be that PAR-CLIP reads are much shorter than common RNA-Seq reads, which reach lengths of 200 bases and show high-quality regions in the middle read length range (Laehnemann, Borkhardt & McHardy, 2015; Schirmer et al., 2015). We used these observations for the design of a PAR-CLIP read simulator that embeds PAR-CLIP specific information within the simulation process and the PARA-suite pipeline for error-aware read alignment and processing. The read simulator mimics PAR-CLIP datasets with error profiles drawn from real PAR-CLIP datasets.

Based on the simulated PAR-CLIP datasets, we determined the parameter settings that delivered the best performance for commonly used aligners (Mukherjee et al., 2011; Ascano et al., 2012b; Sievers et al., 2012; Mukherjee et al., 2014). Our analysis showed that the read alignment is crucial for detecting RBP binding sites in PAR-CLIP datasets. The PAR-CLIP specific read properties make it nearly impossible to identify splice junctions covered by PAR-CLIP reads with RNA-Seq read aligners such as TopHat, STAR or Subjunc, as their algorithms are based on unmet assumptions, such as a similar read coverage across all exons or long reads, in order to achieve high confidence k-mer spectra. Accordingly, these three aligners were outperformed by the other methods (Tables S3–S4). Interestingly, MOSAIK, an error-aware aligner based on hash queries that has been shown to be more robust on RNA-Seq reads than BWT-based aligners (Lee et al., 2014), was also outperformed by most of the other tested methods. Although it was robust on longer RNA-Seq reads, MOSAIK seemed to struggle with the very short PAR-CLIP reads. The PARA-suite alignment pipeline allowed us to increase the fraction of aligned reads in comparison to other aligners, including the alignment of reads spanning exon–exon junctions, both for PAR-CLIP datasets and data from a HITS-CLIP study (Supplemental Information 1). We observed this improvement irrespective of the binding site detection algorithm applied downstream. Importantly, unlike the error-aware short read aligner BWA PSSM, our short read alignment algorithm does not need the manual input of an error profile, which is instead inferred de novo within individual sequencing runs. The aligner thus automatically adapts to varying qualities of individual (PAR-)CLIP sequencing runs and is specifically adjusted to each sequence dataset. To our knowledge, it is the first tool for simultaneous de novo error model inference and short read alignment based on the BWA algorithm. Another difference from the BWA PSSM algorithm is that the latter introduces mismatches while considering the base calling quality scores and a probabilistic background model for matching bases in addition to the input error profile. In contrast, the generic error profile estimation of the PARA-suite is not limited to any specific input profile. Further applications of our software could thus be used to analyze other types of error-prone sequencing data such as bisulphite sequencing data, which introduces a high amount of C–T mutations (Frommer et al., 1992) or data from low-quality ancient DNA samples (Briggs et al., 2007).

Common read simulators such as ART or GemSim do not allow simulating PAR-CLIP reads with their specific error profiles. When comparing the PAR-CLIP read simulator to the recently developed CSeq simulator for CLIP data (Kassuhn, Ohler & Drewe, 2016), both have different strengths. CSeq takes an exact binding motif and T–C conversion profile that is specific for the respective binding motif as input, thus restricting the reads’ base composition and T–C conversion sites. This allows to mimic PAR-CLIP reads for a specific RBP, but not to generalize evaluations on these datasets to all kinds of RBPs. In comparison, the PAR-CLIP reads simulated with the PARA-suite are based on data that have been inferred from three different PAR-CLIP datasets to simulate heterogenic reads, which represent a broader spectrum of RBP binding sites. In addition, the read selection is not restricted to sequences containing the actual RBP binding motif. Thus, CSeq and the PARA-suites’ read simulator have slightly different applications: CSeq allows one to simulate reads to optimize parameters for a specific dataset and the PARA-suite allows one to simulate reads for general tool evaluation and algorithmic improvements.

Our analysis of combinations of read aligners and binding site detection algorithms on simulated and real datasets indicated that no single software performed best in terms of detecting binding sites on the available PAR-CLIP datasets. This observation was recently also made on other datasets (Kassuhn, Ohler & Drewe, 2016). Our analysis of the HuR and FUS datasets revealed that U-rich binding sites tended to show higher rates of T–C conversions per read and were best aligned by BWA PSSM. RBPs with a more heterogeneous nucleotide distribution within the binding site (e.g., EWSR1 and TAF15) are better assessed by BWA PARA. This is supported by an analysis of uridylate-rich sequences from our simulated data aligned by BWA PSSM and BWA PARA (Supplemental Information 1). Therefore, a preliminary analysis of the error profile using the PARA-suite error profiler could allow one to determine the best approach to analyze sequencing data of a novel yet uncharacterized RBP.

Supplemental Information

Supplemental Information 1 Supplementary material, including Figures, Tables and Methods

Click here for additional data file.

The authors thank Johannes Droege, David Laehnemann and Cristina della Beffa for their critical comments on the manuscript.

Additional Information and Declarations

Competing Interests

Author Contributions

Data Availability

Alice C. McHardy is an Academic Editor for PeerJ.

Andreas Kloetgen conceived and designed the experiments, performed the experiments, analyzed the data, contributed reagents/materials/analysis tools, wrote the paper, prepared figures and/or tables, reviewed drafts of the paper.

Arndt Borkhardt and Jessica I. Hoell analyzed the data, wrote the paper, reviewed drafts of the paper.

Alice C. McHardy conceived and designed the experiments, analyzed the data, contributed reagents/materials/analysis tools, wrote the paper, reviewed drafts of the paper.

The following information was supplied regarding data availability:

Source code of the PARA-suite toolkit and the PARA-suite aligner (BWA PARA) are available at https://github.com/akloetgen/PARA-suite and https://github.com/akloetgen/PARA-suite_aligner, respectively, under the GNU GPLv3 license.

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
