# Peer review of "The PARA-suite: PAR-CLIP specific sequence read simulation and processing"

_PeerJ, doi:10.7717/peerj.2619_

## Round 0.1 · original submission · Major Revisions

The manuscript has been carefully reviewed by two external reviewers.

The reviewers appreciated the tools presented here and the methodology behind it. However, they both found that the manuscript will require major efforts and further work before publication. This is especially true with regards to the comparison with different simulation tools of PAR-CLIP’s data already available, better assessing the statistical significance of the results, revising the english form of the manuscript, as well as improving the documentation for the scripts. I would like thus to encourage the authors to address point by point all the precious comments and concerns raised by the reviewer.

Reviewer 1 ·

Basic reporting

There were No Comments for the criteria not explicitly listed below.


"The article must be written in English using clear and unambiguous text and must conform to professional standards of courtesy and expression."

The article is, mostly written in clear and unambigous language. However, several parts of the manuscript do need to be revised for clarity, e.g.:

1.1. The authors define the terms "PARA-suite pipeline" and "PARA-suite aligner" to refer to the full pipeline, and to the modified (by the authors) BWA aligner, respectively (section 3.3, lines 404). However, these definitions are not used consistently. For example, Table 1 lists "PARA-suite pipeline" and "PARA-suite", and in subsequent sections, the authors fall back to using "PARA-suite" to refer to the full pipeline. The authors must take care that terms are used consistently in the entire manuscript! To avoid confusion, it may be worthwhile to refer to the modified version of BWA used in the PARA-suite pipeline as BWA-PARA, similar to how the authors refer to 'BWA 2MM' and 'BWA-PSSM'.

1.2. Does the "Real time(s)" for the PARA-suite pipeline include the time spent by the BWA 2MM mapping used to obtain the error and T-C conversion profiles used by the PARA-suite pipeline? It is not clear from the text, but the numbers (less than BWA 2MM + PARA-suite aligner) suggest not. Please clarify in the text.

1.3. On line 429, the authors comment on the performance of the PARA-suite pipeline and PARA-suite aligner "out of eight analyzed alignment scenarios". It is not clear how this relates to the analyses that the authors have described in this section.

1.4. The PARA-suite aligner and the PARA-suite pipeline seems to be swapped in the sentence on line 427. "Here, the PARA-suite aligner and pipeline", seems that it should be "Here, the PARA-suite pipeline and aligner", to match the results presented in table 1.

1.5. The sentence on line 223, "Dashes boxes show packages while the other ones are executable programs" could use some care; the sentence starting on line 432 is even harder to read.


"The structure of the submitted article should conform to one of the templates ..."

1.6. Line 54 contains what appears to be a garbled reference that was not properly imported into the document (Eddy 2001?). It is further absent from the References list.


"Figures should be relevant to the content of the article, of sufficient resolution, and appropriately described and labeled."

1.7. While a visual comparison between BWA backtrack and the algorithm of the PARA-suite aligner is helpful (Figure 3), the authors introduce this figure before introducing the termination criteria for the PARA-suite aligner, making it largely impossible to follow. Preferably, the figure should be introduced after the difference between BWA backtrack and PARA-suite aligner is described, or include information about their respective termination criteria so that the reader can actually read it. The figure could further be improved by including a legend in the figure itself (rather than writing in the text what color corresponds to what algorithm), and by adding something like arrows to the BWA / PARA-suite aligner paths to indicate their direction.

1.8. Figure S2 A-B: Please use the same scale on the Y-axis, as to allow direct comparison of error rates.

Experimental design

There were No Comments for the criteria not explicitly listed below.

"The investigation must have been conducted rigorously and to a high technical standard."

2.1. For the analyses of FET PAR-CLIP datasets (section 3.4, Table 2), the authors uses a restricted selection of alignment algorithms. Why, and how where these selected? Similarly, why the selection of mappers used for the analyses of the HuR binding site data (section 3.5).

2.2. Base quality scores are randomly assigned to nucleotides based purely on the position in the read, in reads simulated by the PAR-CLIP read simulator (lines 196-197 and createSimulatedPARCLIPDataset.pl). As such, an assigned quality score would not be expected to be informative about the actual quality of a given simulated base-call. However, BWA-PSSM explicitly incorporates the base quality scores into the PSSM calculation during mapping (Kerpedjiev et al 2014, and as the authors themselves note on lines 562-4), and I would therefore expect this tool to perform sub-optimally given this type of simulated data. The authors should address this in the text, given that they compare the performance of this tool with their PARA-suite aligner.


"Methods should be described with sufficient information to be reproducible by another investigator."

2.3. Please note parameters used by programs where applicable, even if just to note that default parameters are used. For example, how was cutadapt run?

Validity of the findings

No comments.

Additional comments

4.1. While the authors must be commended for including an example project with their pipeline, it is broken. Firstly, the file refers to the jar file 'parma.jar', while the included jar is named 'parasuite.jar'. The script further fails at multiple stages even the name of the jar is corrected, seemingly due to changes to the command-line interface for parasuite (e.g. attempts to call 'parasuite.jar error' with the option -q, which is not accepted by this command), and due to missing files (which may be caused by earlier errors).

4.2. Build instructions for the PARA-suite toolkit would be appreciated.

4.3. In the text for figure 3, the authors further claim that their algorithm makes 14 comparisons vs 16 for BWA backtrack and that "therefore, the PARA-suite is slightly faster than BWA". However, the authors do not justify this claim for the general case, making this a (pointless) statement about a single, hypothetical alignment. The authors will either need to argue that their algorithm has an *expected* lower runtime in the *general* case, or draw their conclusions with regards to the runtime of the PARA-suite aligner from their actual results.

4.4. On line 328 and line 338, the authors describe their trimmed PAR-CLIP reads as "shorter than 30 bases", and as having "a length usually shorter than 30 bases". That is not very informative; please include the mean or median length.

4.5. Why is the results from the analysis of PAR-CLIP data on HuR (section 3.5) not included in the main text, when the corresponding results for FET are (section 3.4)?

4.6. In Table S3, the authors list several programs as "Name", "Name X1", "Name X2", and "Name X3". It is unclear what this refers to, and I could not find any explanation in the text.

4.7. (Minor) Use a single 3x2 matrix in the formula on line 270 (rather than two 3x1 matrices), to ensure that the descriptions line up with the corresponding formula.

·

Basic reporting

1.1 The submission must adhere to all PeerJ policies.
The article meets PeerJ’s standards and policies.

1.2 The article must be written in English using clear and unambiguous text and must conform to professional standards of courtesy and expression.

There are few grammatical and typing errors throughout the text as well. A good idea would be that a native English speaker edits the paper for grammar and other minor issues.

1.3 The article should include sufficient introduction and background to demonstrate how the work fits into the broader field of knowledge. Relevant prior literature should be appropriately referenced.

I would like to invite authors to consider some recent literature on this evolving and dynamic technology, for example:
Danan, PAR-CLIP: A Method for Transcriptome-Wide Identification of RNA Binding Protein Interaction Sites, 2016. http://www.ncbi.nlm.nih.gov/pubmed/26463383
In addition it should be an adding value to address a comparison with different methodologies, for this reason I suggest the authors to read some related review:
Wang, Design and bioinformatics analysis of genome-wide CLIP experiments, 2015
http://nar.oxfordjournals.org/content/early/2015/05/09/nar.gkv439.full
The advent of cross-linking immunoprecipitation coupled with high-throughput sequencing (genome-wide CLIP) technology has recently enabled the investigation of genome-wide RBP–RNA binding at single base-pair resolution. This technology has evolved through the development of three distinct versions: HITS-CLIP, PAR-CLIP and iCLIP.
Interesting comparison:
http://epigenie.com/quick-review-crosslinking-immunoprecipitation-clip-methods/

1.4 The structure of the submitted article should conform to one of the templates. Significant departures in structure should be made only if they significantly improve clarity or conform to a discipline-specific custom.
The article meets PeerJ’s standards and policies.



1.5 Figures should be relevant to the content of the article, of sufficient resolution, and appropriately described and labeled.

I noted low-resolution figures within the text conforming the structure and the template of PeerJ on first submission, please let me ask the authors to prepare individual figure files in one of the following formats: PowerPoint (.ppt), Tagged Image File Format (.tif), Encapsulated PostScript (.eps), Adobe Illustrator (.ai) (please save your files in IIllustrator's EPS format), Portable Network Graphics (.png), or editable PDF. The resolution must be at a minimum resolution of 600 d.p.i. for line drawings (black and white) and 300 d.p.i. for colour or greyscale.

1.6 The submission should be ‘self-contained,’ should represent an appropriate ‘unit of publication’, and should include all results relevant to the hypothesis. Coherent bodies of work should not be inappropriately subdivided merely to increase publication count.
No comments.

1.7 All appropriate raw data has been made available in accordance with our Data Sharing policy.
No comments.

Experimental design

2.1 The submission must describe original primary research within the Scope of the journal.
The manuscript appears with an implementation of set of freely available tools analyzing and processing PAR-CLIP’s data. In particular the authors included error model inference, PAR-CLIP read simulation based on PAR-CLIP specific properties, a full read alignment pipeline with a modified Burrows-Wheeler Aligner (BWA) algorithm and CLIP read clustering for binding site detection. The simulation’s part sounds interesting and novel, but I invite the author to consider a comparison related to different simulation tool of PAR-CLIP’s data from literature.
I can suggest for example:
https://genomebiology.biomedcentral.com/articles/10.1186/gb-2014-15-1-r18#MOESM1
http://www.ncbi.nlm.nih.gov/pubmed/26463385

In addition I would like to see differences and adding-value of the proposed methodology (PAR-CLIP simulator) considering for example
Kassuhn W1, Ohler U, Drewe P., CSEQ-SIMULATOR: A DATA SIMULATOR FOR CLIP-SEQ EXPERIMENTS., Pac Symp Biocomput. 2016;21:433-44.
http://www.ncbi.nlm.nih.gov/pubmed/26776207

It would be interesting to see comparison with performance evaluation of different PAR-CLIP’s data simulator tools where in this context it would be useful and critical that those tools should have the same characteristics as real datasets.
2.2 The submission should clearly define the research question, which must be relevant and meaningful. The knowledge gap being investigated should be identified, and statements should be made as to how the study contributes to filling that gap.
I think it should be more useful for readers if the authors can explain in more details their research question. For some points it is not clear if the manuscript’s message deals with a suite of different PAR-CLIP data analysis tools or with PAR-CLIP’s read simulator. It is just a clearly definition of the message, the authors structured their abstract in sub-sections Background, Methods, Results, Availability they could define their aim, inspiring for example Cseq-Simulator’s abstract:

“It has not been assessed which of the available tools are most appropriate for the analysis of CLIP-Seq data. This is because an experimental gold standard dataset on which methods can be accessed and compared, is still not available. To address this lack of a gold-standard dataset, we here present Cseq-Simulator”

2.3 The investigation must have been conducted rigorously and to a high technical standard.
No comments.

2.4 Methods should be described with sufficient information to be reproducible by another investigator.
The authors provided two github repositories such as:
https://github.com/akloetgen/PARA-suite
https://github.com/akloetgen/PARA-suite_aligner
In particular it would be interesting and useful in the context of reproducibility to add supplementary sections within the manual and the examples providing codes, parameters and figures to reproduce the results, reported in the Table1, in the PeerJ’s manuscript.
https://github.com/akloetgen/PARA-suite/blob/master/Manual.pdf
https://github.com/akloetgen/PARA-suite/tree/master/examples

2.5 The research must have been conducted in conformity with the prevailing ethical standards in the field.
The article meets PeerJ’s standards and policies.

Validity of the findings

3. VALIDITY OF THE FINDINGS

3.1 The data should be robust, statistically sound, and controlled.
Robust
I would like to see if the performances of proposed methodology such as Para-suite are influenced on data re-sampling and bootstrapping. It could be interesting to apply a cross-validation procedure and fitting the model on training data and then predict the results on testing data. In that context I would consider robust results.

Statistically sound:
In the supplementary table S1, they dind’t reported any pvalues or corrected FDR.
I would like to see differences (increasing FDR) applying any statistical hypothesis testing, for example Fisher’s exact test (FET) that authors prefer.
Without any FDR how can they assess the significance of their results in table S1?
They should also provide the same, a FET for the venn diagram in Supplementary Figure S6.
See for example table1 and table 2 of Chen,PIPE-CLIP: a comprehensive online tool for CLIP-seq data analysis, Genome Biology, 2014
https://genomebiology.biomedcentral.com/articles/10.1186/gb-2014-15-1-r18#Tab2
https://genomebiology.biomedcentral.com/articles/10.1186/gb-2014-15-1-r18#Tab1

‘Controlled’
I have some concerns regarding supplementary tableS4, please the authors can double-check for which methodologies they reported TN and FN equal to zero, because it sounds strange.
If their findings are consistent, please the authors can provide further explanations.

3.2 The data on which the conclusions are based must be provided or made available in an acceptable discipline-specific repository.
See section Experimental design 2.4 for repository methodologies. In addition data used for method section 2.1 of the manuscript, should be summarized in a table reporting platforms (PAR-CLIP, HITS-CLIP, etc), samples, experiments, datafile, accession numbers, etc.

3.3 The conclusions should be appropriately stated, should be connected to the original question investigated, and should be limited to those supported by the results.
In the discussion section (row 530-531) the authors asserted that “We characterized some of the unique properties of PAR-CLIP sequence datasets that have, to our knowledge, so far not been analyzed, such as preferred read positions for T–C conversion sites and their frequencies per read position.” This point is interesting but not original, can the authors consider novel CSEQ-simulator findings, as I mentioned in section 2.1 and dealing with the results comparison. I reported here an extract of CSEQ-simulator’s manuscript:


“Finally, in order to generate the CLIP-Seq reads, we induce the diagnostic events (e.g. T-C conversions, deletions and truncations) in the raw reads. To this end, we sample the diagnostic
events in the reads according to user specified distribution (diagnostic event profile) that is
centred on the binding site.” From Kassuhn W1, Ohler U, Drewe P., CSEQ-SIMULATOR: A DATA SIMULATOR FOR CLIP-SEQ EXPERIMENTS., Pac Symp Biocomput. 2016;21:433-44.


3.4 Speculation is welcomed, but should be identified as such.
No comments.

3.5 Decisions are not made based on any subjective determination of impact, degree of advance, novelty, being of interest to only a niche audience, etc. Replication experiments are encouraged (provided the rationale for the replication, and how it adds value to the literature, is clearly described); however, we do not allow the ‘pointless’ repetition of well known, widely accepted results.
I have a concern related to methodologies used for read aligners.
I invite the authors to provide further proofs as a reference a table for example, from where they can show why they selected Bowtie, Bowtie2, BWA PSSM, STAR, etc.
(http://www.hindawi.com/journals/bmri/2014/309650/tab1/)
The reason could be dual-fold: (i) in terms of fitting approaches with different performances depending on different data-type (i) approached mostly used and cited as well as reduced computation time for example [http://www.hindawi.com/journals/bmri/2014/309650/fig2/#a] .

3.6 Negative / inconclusive results are acceptable.
No comments.

---

## Round 0.2 · Minor Revisions

The manuscript can be accepted upon the minor revisions suggested by the reviewer. Please, pay especially attention to Table S3 and to the pipeline-related errors. Indeed, you should review the error handling in their pipeline and ensures that the program correctly detects and reports failures.

Reviewer 1 ·

Basic reporting

The authors have adequately addressed my concerns, and I have no additional comments following their revisions.

Experimental design

The authors have adequately addressed my concerns, and I have no additional comments following their revisions.

Validity of the findings

No comments.

Additional comments

For most part, the authors have adequately addressed my comments. However, I have a clarification of point 4.6, and additional comments following their addressing of 4.1.

My original point 4.6 was as follows, "In Table S3, the authors list several programs as "Name", "Name X1", "Name X2", and "Name X3". It is unclear what this refers to, and I could not find any explanation in the text". I apologize for being unclear, as it the authors took this to mean literal instances of "Name X1", "Name X2", etc. in the table. My question was in fact in relation to the names used in table S3, such as the following:

- PARAsuite pipeline
- PARAsuite X1 pipeline
- PARAsuite X2 pipeline
- PARAsuite X3 pipeline

It is still not clear to me what those refer to, and the authors should probably clarify this in the legend.


Secondly, I thank the authors for adding a build file for their the PARA suite pipeline and for updating their example project script. I was successfully able to both build the jar and to run the example project included with the PARA suite pipeline.

However, getting this example project to run also revealed to me that the pipeline currently does a terrible job of handling errors while running the pipeline. For example, the pipeline does not check if calling the simulator succededed, and will happily proceed even if that program is missing or broken:
https://github.com/akloetgen/PARA-suite/blob/master/src/src/main/Main.java#L719

The return-code of such 'waitFor' calls must be checked, and the pipeline itself should return a non-zero return-code upon detecting such failures.

Similarly, a large number of program invocations via the 'executeCommand' function are unchecked (it returns the result of a 'waitFor' call). I note that all cases where the 'executeCommand' calls are checked follow the same form (namely raising a MappingErrorException with the command-line arguments), and would therefore suggest simply moving this error-checking into the executeCommand to ensure that call program invocations are checked.

Finally, the PARA suite pipeline will catch exceptions in a number of cases, manually print a stack-trace, and then either invoke 'System.exit(0)' or do nothing. This has the effect of making automated failure detection impossible, even in the cases where System.exit is actually called, as return-code of 0 is meant to indicate success.

See for example
https://github.com/akloetgen/PARA-suite/blob/master/src/src/main/Main.java#L734-L740
https://github.com/akloetgen/PARA-suite/blob/master/src/src/main/Main.java#L436
out of many more instances.

Missing System.exit calls must be added, and an appropriate (non-zero) return-code must be used in all System.exit calls. In a number of cases, simply removing the call to 'e.printStackTrace()' and just re-throwing the exception should suffice, since the JVM will print a stack-trace for unhandled exceptions, and return an appropriate return-code when this happens.


While I do not consider this reason for rejecting the paper, this would preclude my use of the PARA suite pipeline for anything but trivial data-sets where I could run the pipeline by hand and manually inspect the (verbose) output from every invocation of the tool. This, however, does not scale for larger data sets, and significantly raises the bar for the PARA suite pipeline being integrated into automated workflows.

I would therefore strongly recommend that the authors review the error handling in their pipeline, and ensures that the program correctly detects and reports failures, as this would improve not just the reliability of their tool, but also it's usability in multiple contexts.

---

## Round 0.3 · accepted · Accept

All the previous concerns have been nicely addressed and I can finally endorse this interesting manuscript for publication.